# The Effect of COVID-19 on the Hospitality Industry: The Implication for Open Innovation

**Kanwal Iqbal Khan** [1] , **Amna Niazi** [2,*]**, Adeel Nasir** [3,*]**, Mujahid Hussain** [4] **and Maryam Iqbal Khan** [5]

1    Institute of Business & Management, University of Engineering and Technology, Lahore 39161, Pakistan;
     drkanwaliqbalkhan@gmail.com
2    Humanities and Social Science Department, University of Engineering and Technology,
     Lahore 39161, Pakistan
3    Department of Management Sciences, Lahore College for Women University, Lahore 54000, Pakistan
4    FAST School of Management, National University of Computer & Emerging Sciences, Lahore 54000, Pakistan;
     mujahid.hussain@nu.edu.pk
5    Institute of Technology and Management, University of the Punjab Lahore Campus, Lahore 54000, Pakistan;
     maryumkhantqm@gmail.com
*    Correspondence: amna.niazi@uet.edu.pk (A.N.); adeel.nasir@lcwu.edu.pk (A.N.); Tel.: +92-3004358679 (A.N.);
     +92-3344074441 (A.N.)

**Abstract:** The current coronavirus pandemic (COVID-19) has led the world toward severe socio-economic crisis and psychological distress. It has severely hit the economy; but the service sector, particularly the hospitality industry, is hard hit by it. It increases the sense of insecurity among the employees and their perception of being unemployed, adversely affecting their mental health. This research aims to contribute to the emerging debate by investigating the effect of economic crisis and non-employability on employees' mental health through perceived job insecurity under the pandemic situation. It empirically examines the underlying framework by surveying 372 employees of the hospitality industry during COVID-19. Results indicate that perceived job insecurity mediates the relationship of fear of economic crisis, non-employability, and mental health. Furthermore, the contingency of fear of COVID-19 strengthens the indirect relationship of fear of economic crisis on mental health through perceived job insecurity. The findings will provide a new dimension to the managers to deal with the psychological factors associated with the employees' mental health and add to the emerging literature of behavioral sciences. The study also highlights the increasing need for investment in the digital infrastructure and smart technologies for the hospitality industry.

**Keywords:** fear of economic crisis; fear of COVID-19; perceived job insecurity; non-employability; mental health; hospitality industry; COVID-19; digital infrastructure

## 1. Introduction

The world is always facing challenges due to technological advancements, natural disasters, and demographic factors. However, the coronavirus (COVID-19) pandemic has emerged into the biggest challenge of the decade. It has affected the peoples' lifestyle and had an inverse impact on their health, social, and financial conditions [1]. Coronavirus is a contagious disease caused by a virus named SAR-CoV-2 (severe acute respiratory syndrome Coronavirus 2) [2]. COVID-19 has been declared as a pandemic by the World Health Organization (WHO) [3]. Coronavirus transmission occurs in droplets generated through the sneezing or coughing of an infected person [4]. In December 2019, the novel COVID-19 (coronavirus disease) appeared for the first time in Wuhan, China. Within months, many countries were affected by it, and the number of patients increased drastically [5]. At the initial stage of this pandemic, different countries managed to control the virus, but still, there are no estimations that soon how many of the nations will be affected due to the second wave of COVID-19 [6].

Since the outbreak of COVID-19, it has become difficult for the health practitioners to deal with it. The mortality rate is high in many countries; thus, this disease is one of its kind with no cure found and confirmed at the time this paper was written [7]. A worldwide recession, economic downturns, and decline in returns of the industries are just some of the consequences of this global emergency [8]. The increasing number of cases has created panic, stress, and nervousness among the masses around the world [5]. People had to encounter psychological issues, panic attacks, anxiety, and understanding that there is no known cure for this disease [9]. The steps taken by the government to counter the fear of COVID-19 include awareness about the disease, information about the precautionary measures, and lockdowns [1]. However, the psychological aspects directly related to the people's mental health were ignored by the authorities. Researchers such as Gallie [10] illustrated that this disease adversely impacts the mental health of the people, leading to anxiety and depression.

Prior studies considered fear of job loss and financial insecurity as the most substantial consequences of governmental policies such as lockdowns [11]. In the private sector, the fear of job insecurity is an ongoing threat; however, it has gained more attention during this pandemic. The fear of being laid off from their ongoing jobs is higher than the fear of being infected. Studies showed that people panic at an individual level because of the threat of losing their income and employment [9]. According to research, it takes years to recover from the fear of non-employability. Fear of COVID-19 has emerged as a global phenomenon. It has impacted individuals, institutions, societies, and nations all across the world. Many businesses failed to survive the economic pressures posed by COVID-19. Those that are surviving had to find and implement innovative business models. Despite such innovations, many organizations reported significant losses leading to downsizing and other cost-cutting mechanisms.

Past research on pandemics suggests that such diseases significantly affect employee performance and mental health. Occupation uncertainty and the threat of unemployment at the workplace has been directly related to an unexpected low performance level [12]. This uncertainty and threat increase anxiety, fear, depression, and job burnout among the employees [13]. Studies revealed that employees' impaired mental health affects their attitude and influences the quality of service the employees provide. Therefore, the management needs to consider mental health of the workers as an important issue during the pandemic [14].

The study conducted by Ramelli and Wagner [15] stated that the world had not faced an economic crisis as bad as that during the COVID-19 pandemic. Almost all the corporate sectors were adversely affected, whereas the first one hit by this was the hospitality sector. Internationally, the hospitality industry is a flourishing sector. In Pakistan, it adds a significant amount to the GDP each year and plays a vital role in the service sector. As in other countries, this sector was also affected adversely in Pakistan. Tourism was at a halt, hotels were not allowed to entertain guests, restaurants were closed for dine-ins, employees were being laid off, and the remaining employees were under the threat of non-employability. This pandemic brought a challenge for the hospitality industry to survive by adopting the innovative strategies and improving the customers' perception of safety [16].

Shin and Kang [17] purposed that by implementing technological innovations and risk reduction strategies, the hospitality industry can gain the trust back of their customers once the restrictions are removed. In addition to the economic crisis, this pandemic created a wave of job insecurity that led toward mental issues in the employees. In their study, Kang, Li [18] stated that mental health holds vital importance for employees to function properly at the workplace. They also further stated that families and friends are also affected by employees who face mental issues.

Given the uncertainty of COVID-19, the current study investigated the potential effect of fear of economic crisis and non-employability on the mental health of the employees of the hospitality industry. The target population for this research was the employees of

the hospitality industry of Pakistan. Since this sector is under-researched, thus, this study tries to fill the gap and open new horizons for the managers. Specifically, in this research, we investigate how perceived job insecurity meditates the relationship between fear of economic crisis, non-employability, and mental health. Finally, this study also comprehensively discusses how fear of COVID-19 moderates the relationship between fear of economic crisis and perceived job insecurity and even between non-employability and perceived job insecurity.

By examining these relationships, we can better understand the factors causing mental health problems among the employees in this pandemic. Therefore, this research would contribute toward the literature by testing the relationships and the effect of the micro and macro-level factors on the employee's mental health. This study also takes the opportunity to know more about the psychological factors impacting the employees during the pandemic. The main objective of the research was to contribute toward the literature of the hospitality sector of Pakistan, since it is a sector that has received little attention from researchers. Secondly, it is one of the industries that is facing significant repercussions by this pandemic. The present paper explores the relationships of fear of economic crisis, non-employability, perceived job insecurity, and mental health issues that employees have to face.

## 2. Review of Literature and Hypothesis Development

### 2.1. Fear of the Economic Crisis and Mental Health

Fear has many implications. Its purpose may include it as a strategy to attain a goal. The fear of the unknown is a driving force to take advantage of the insecurities of the other party. The factor fear has not only been considered as a strategy to motivate employees in corporate settings, but it is also used by political parties to set their agendas [18]. Fear arises from an unplanned situation; in some cases, we can predict the outcome, and in other cases, the results may stay unrevealed. Sudden changes in the socio-structural circumstances lead to anxiety and fear among people. Research reveals that these fluctuations create angst among the masses and compel people to take harsh decisions such as self-harm and suicide [19]. Research indicates that the economic crisis includes unemployment, financial sufferings of employees, and job insecurity [20]. Giorgi [21] also explained the economic crisis as a macro stressor including multiple factors of an employee's economic life, including the fear of job loss and job insecurity.

The literature also considers the fear of economic crisis as an innovative construct and defines it as a perception of the employee about the organization that something such as downsizing will take place in the organization soon [22]. A study revealed that the primary reason for suicide was the pressure that there are no jobs in the market and unemployment. Prolonged economic crisis leads to financial hardships among the people working or trying to find work. These sufferings cause psychological distress and fear of job loss among the employees. The economic crisis has a significant impact on working people [23]. This perception of crisis may shape stress, anxiety, and turnover, and absenteeism may significantly affect the employees' health [24]. Stress theory explains individuals and families' reactions and how they react when they face stress [25]. Although the economic crisis affects everyone, people have a different response and handle stress in varied ways.

The economic crisis due to COVID-19 has raised challenges for the economies to reach back on the track of progress [16]. Every country has been affected by this disease, and not only developmental programs have been at a halt, but people are also struggling mentally and financially. Roca [26] and Voydanoff and Donnelly [27] in their research proved that the economic crisis had a direct and negative impact on the mental health of employees. Previous research also illustrated that employees' mental health is adversely affected by the economic crisis prevailing in the country [28]. The study conducted by Voydanoff [29] also highlighted that income loss, lack of finances, and unemployment also causes depression and affect the employees' mental health.

## 2.2. Non-Employability and Mental Health

Perceived non-employability is a susceptible issue for the employees. If employees perceive that other organizations are laying off their employees due to the prevailing situation, they become even more sensitive. This phenomenon is also explained by the uncertainty management theory [30]. This theory describes that during uncertain times, employees become more sensitive and vulnerable. Organizations need to be more considerate, since the employees become exposed to the treatment they receive from their organization. Employees who fear unemployability consider their organizations to be more non-cooperative, and hence, they have a higher level of fear of the economic crisis [20]. Berntson and Marklund [31] define employability as the set of working skills that would find employees another job. It is considered to be the perception made in the mind of the employees that would find them a new job when needed in the market.

A worker perceiving non-employability would believe that it would be difficult and sometimes impossible to find a job with the skills set that he has [21]. This thought and perception of not getting a new job consume the psychological resources of the employee, hence impacting mental health, especially during austere times. The perceived fear of non-employability adversely affect the employees; even their families suffer due to this perception [32]. Low-income families are more affected by fear of non-employability, resulting in adverse mental and health issues [33]. Previous research has already established that non-employability is associated with mental health problems such as depression, anxiety, and even self-harm [34]. Giorgi [21] also stated that the psychiatric cases and even mortality rate due to psycho-pathological conditions increased during the economic crisis.

## 2.3. Mediating Role of Perceived Job Insecurity

Job insecurity is defined as the fear that one will lose their job and will not be able to retain the job [35]. Greenhalgh and Rosenblatt [36] further added to the definition of job insecurity as the vulnerability that one faces while working in an organization with a threat of unemployment. Previous research has proved that job insecurity has many implications. Some of the significant repercussions to job insecurity include complaints regarding mental health, lack of commitment, and low job satisfaction [37]. The economic crisis may be considered one of the most triggering factors that creates fear of job insecurity. Due to the economic instability and fear of job loss, employees have also developed a state of mind where they believe that long-term relationships with organizations are no longer considered loyal [38]. According to Elman and O'Rand [39], employees have also generated reasoning that no one is resistant to job insecurity. The sense of vulnerability among employees augments the fear of job insecurity, and this phenomenon affects the mental health of the employees [36].

The literature is filled with evidence that proves a negative relationship between employability and perceived job insecurity [40]. Silla [41] highlighted that job insecurity is negatively associated with an employee's well-being. Otterbach and Sousa-Poza [42] also supported a negative relationship between job insecurity and mental health of employees. However, the present research explores the missing link by taking job insecurity as a mediator among non-employability and mental health. Job insecurity is one of the significant factors that cause restlessness among the employees, hence impacting the mental health of the employees directly.

Research conducted by Llosa [43] also concluded the significant effects of job insecurity on the mental health of employees. Their study highlighted that employees' professional networking, low satisfaction with life, and other mental and psychological issues are standard among employees who feel that their job is not secure. A study by Gallie [10] stated that a major reason why an employee fears job insecurity is the awareness in the reduction in staff by the employers. Hence, the present research combines the macro-level factor, i.e., economic crisis and macro-level organizational concepts, and tries to explain the impact of fear of economic crisis and non-employability on the employees' mental

health [35]. We cannot ignore the importance of mental health of employees, since it directly affects organizational performance [44]. Thus, we hypothesize the following:

**Hypothesis 1a (H1a).** *Fear of economic crisis has a significant negative impact on the employees' mental health through the mediation of perceived job insecurity.*

**Hypothesis 1b (H1b).** *Non-employability has a significant negative association with the employee's mental health through the mediation of perceived job insecurity.*

*2.4. Moderating Role of Fear of COVID-19*

Wars have boundaries; however, diseases such as COVID-19 proved no boundaries when there is an outbreak. Contagious diseases such as COVID-19 leave a trauma on people's mental health, since they affect the well-being of people [45]. Fear is an expressive response to this situation. Fear of COVID-19 created panic, excessive stress, and anxiety among the employees that lead toward job insecurity. People, especially the working class, felt vulnerable and dealt with psychological issues because of uncertainty [46]. Scholars believe that the fear of COVID-19 increases the fear of economic crisis that adversely affects employees' mental health [47]. The fear of COVID-19 increased drastically among the employees because of uncertainty, job insecurity, financial insecurities, and governmental health care policies.

The unprecedented measures taken by the government to control the outbreak itself created panic and triggered mental issues (Galea et al. 2020). Bao [48] also endorsed in their research that disease such as COVID-19 is positively linked with mental stress and anger. Researchers also confirmed that the impact of such diseases does not remain with the affected persons only but also spreads out to families, communities, and nations [49]. Throughout the world, due to COVID-19, schools and universities were closed, large gatherings were prohibited, workers were being laid off by organizations, non-employability increased, and the economy faced the worst crisis. This all had a direct impact on the employee's mental health.

Several studies report the amount of research being conducted on making the vaccine for COVID-19. However, hardly any study shows how the employees mentally suffered due to this pandemic. Employees were directly affected, since the functioning of businesses changed. Supply chains were broken due to the non-availability of raw materials owing to travel restrictions. When the manufacturing and selling were negatively affected, organizations had to cut down on costs, and hence, firing employees was one of the strategies to reduce cost. Reducing the workforce involved the employees who were being laid off and created a sense of job insecurity among the remaining employees [50]. Job insecurity shatters the employee's confidence and hence leads to a mental health problem for the employee. Therefore, this study highlights that employers had to bear a loss in revenue during this period and facilitate the employees to recover from psychological stress. Thus, the following hypotheses are proposed:

**Hypothesis 2a (H2a).** *Fear of COVID-19 moderates the indirect relationship of fear of economic crisis on mental health, such that the effect of fear of economic crisis on mental health through perceived job insecurity might be more assertive when employees have higher fear of COVID-19.*

**Hypothesis 2b (H2b).** *Fear of COVID-19 moderates the indirect relationship of non-employability on mental health, such that the effect of non-employability on mental health through perceived job insecurity might be more robust when employees have higher fear of COVID-19.*

Figure 1 explains the hypothesized model of the research.

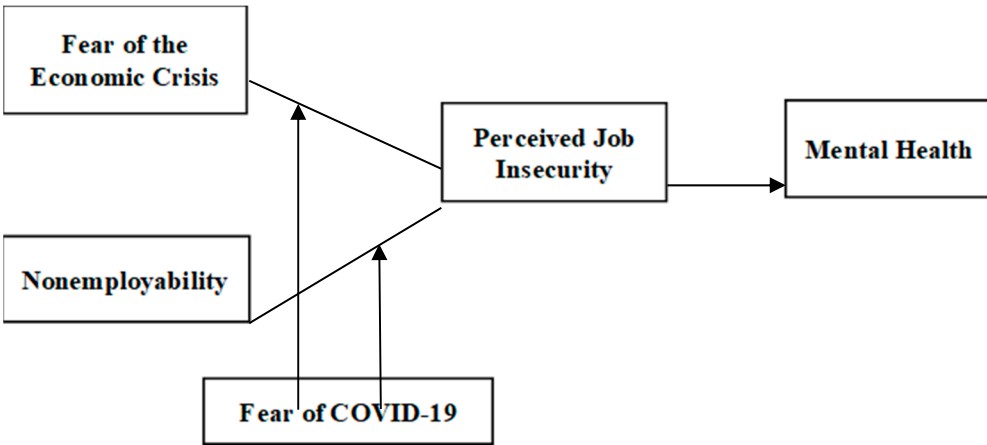

**Figure 1.** Research Model.

## 3. Materials and Methods

### 3.1. Participants and Procedure

The service sector is hard hit by the COVID-19 pandemic, particularly the hospitality industry, based on the accommodation and food and beverages segments. One way to control the spread of COVID-19 was the lockdown, which was the main reason for the cancellation of economic activities. The hospitality industry was also affected by it as national and international flights were halted, and there was no tourism, business trips, meetings, and no flight crew staying at the airport hotels. Furthermore, hotel operations such as restaurants, conferences, seminars, and banquets were also restricted, severely affecting economic and financial performance. However, earlier to this pandemic, the hospitality industry was growing and flourishing in Pakistan. Its contribution to GDP was 3%, and from 2012 to 2014, the industry's growth rate was observed as 7% [51], which was increasing year by year. Due to this worst-case scenario, the hospitality industry had lost approximately $253.7 million and just $0.64 million in February. It was also reported that 90% of hotel bookings were cancelled in this period [51], which adversely affected their profitability and created chaos among the employees regarding their future employability.

That is why the current study targeted the hospitality industry employees to analyze the effect of fear of economic crisis and non-employability on mental health through perceived job insecurity. It also explores the moderating role of fear of COVID-19 between fear of economic crisis, non-employability, and mental health of employees. Data collection was initiated during the period of the COVID-19 pandemic (May to July 2020). We have taken prior approval from the institutional ethical committee before initiating the research. The management and employees were also informed about it. Furthermore, the study participants were ensured that the gathered information would not be shared with any authority at any point in time. All the information would be used for research purposes only. We have contacted the administration of 50 hotels and requested them to provide the contact details of their employees (emails and phone numbers) during the smart lockdown period, but only 27 agreed to share the information.

The data were collected from the employees through an online survey. Questionnaires were distributed to the targeted sample through online platforms. We had sent questionnaire links to 750 employees, but 443 participants had filled the questionnaire that presented the response rate of 59%. Initially, the data were screened for missing values, unengaged responses, and multivariate outliers. We further deleted 71 responses and ended up with the final data of 372 participants. The respondent's characteristics are reported in Table 1 that presented the mean (median) age of the participants 31.5 (30) and SD (3.60). Off all the participants, 27 were managers, 148 were engaged in the white-collar jobs, and 197 were performing the blue-collar jobs. A large number of employees (197) were working on fixed-term job contract positions. The mean(median) values related to their length of service or job experience in the contemporary hotel were between 8.5 (10.5) in their

current organization. Overall, 216 employees were male, 249 were married, and most of the employees have obtained their postgraduate degree (187). The detailed characteristics of the respondents are explained in Table 1.

**Table 1.** Socio-demographic data (n = 372).

| Variables | n = 372 |
|---|---|
| **Gender, n (%)** | |
| Female | 135 (36%) |
| Male | 216 (58%) |
| Prefer not to say | 21 (6%) |
| **Marital Status, n (%)** | |
| Married | 249 (67%) |
| Unmarried | 119 (32%) |
| Divorced/Widowed | 4 (0.1%) |
| **Age (years)** | |
| Mean (SD) | 31.5 (3.60) |
| Median | 30 |
| Range | 21–47 |
| **Education, n (%)** | |
| High School Education | 36 (10%) |
| Bachelor Degree | 120 (32%) |
| Postgraduate Degree | 187 (50%) |
| Professional Level Diploma or Course | 29 (0.8%) |
| **Job Position, n (%)** | |
| Managers/Middle Managers | 27 (7%) |
| White-Collars | 148 (40%) |
| Blue Collars | 197 (53%) |
| **Job Seniority (years)** | |
| Mean (SD) | 8.5 (4.20) |
| Median | 10.5 |
| Range | 0.3 (months)–25 (years) |
| **Job Status, n (%)** | |
| Permanent | 175 (47%) |
| Fixed-Term Job Contract | 197 (53%) |

*3.2. Measures*

Fear of economic crisis measures employees' perceived degree that their organizations would be affected by the economic crisis [52]. It was calculated by five questions taken from Giorgi [22]. Non-employability is the perceived degree of employees about their working competencies that do not allow them to acquire another job. It was assessed by 5 items borrowed from Giorgi [53]. After recoding negatively worded items, the highest score indicates each stressors' greater extent: fear of economic crisis and non-employability. The reliability values of fear of economic crisis and non-employability are 0.906 and 0.925, respectively. Perceived job insecurity is the perception of uncertainty that is in the mind of employees regarding the future of their jobs. It was measured by following the way of Kinnunen [54] by 7 items. The highest score of perceived job insecurity represented the greater extent of job insecurity among the employees. Its factor loading lies between 0.719 and 0.901, and the reliability score is 0.922.

Fear of COVID-19 depicts a person's anxiety or fear about the COVID-19 pandemic [55]. It was measured using Reznik [56] and Satici [57] through 7 items. Its highest score indicated the greater level of fear about the coronavirus among the masses, while the lowest score reflected the presence of a lower degree of fear about the pandemic. The reliability of the scale is 0.941. Mental health includes anxiety and depression, social dysfunction, and loss of confidence [52]. It was measured by 12 items from the General Health Questionnaire (GHQ-12), which was designed to diagnose the psychiatric disorder among the people [58]. It was a self-reported questionnaire that described individuals' behavior and symptoms, if any of them, they recently experienced related to their psychological health. It consisted of

12 items: 6 for anxiety and depression, 4 for social dysfunction, and 2 for loss of confidence. It includes both positive and negative questions. The factor loadings of the items range from 0.810 to 0.902. The internal consistency and reliability of the scale are also accepting all the items; for further detail, see Table 2.

**Table 2.** Measurement items and standardized factor loadings.

| Constructs | SFL |
|:---:|:---:|
| **Mental Health (MH)** | |
| α = 0.967; CR = 0.971; AVE = 0.734 | |
| MH 1 | 0.835 |
| MH 2 | 0.834 |
| MH 3 | 0.834 |
| MH 4 | 0.849 |
| MH 5 | 0.863 |
| MH 6 | 0.842 |
| MH 7 | 0.810 |
| MH 8 | 0.858 |
| MH 9 | 0.902 |
| MH 10 | 0.875 |
| MH 11 | 0.888 |
| MH 12 | 0.887 |
| **Fear of Economic Crisis (FEC)** | |
| α = 0.906; CR = 0.930; AVE = 0.727 | |
| FEC 1 | 0.812 |
| FEC 2 | 0.808 |
| FEC 3 | 0.880 |
| FEC 4 | 0.877 |
| FEC 5 | 0.884 |
| **Non-Employability (NE)** | |
| α = 0.925; CR = 0.944; AVE = 0.770 | |
| NE 1 | 0.828 |
| NE 2 | 0.907 |
| NE 3 | 0.901 |
| NE 4 | 0.899 |
| NE 5 | 0.850 |
| **Perceived Job Insecurity (PJI)** | |
| α = 0.922; CR = 0.938; AVE = 0.686 | |
| PJI 1 | 0.826 |
| PJI 2 | 0.767 |
| PJI 3 | 0.844 |
| PJI 4 | 0.719 |
| PJI 5 | 0.856 |
| PJI 6 | 0.866 |
| PJI 7 | 0.901 |
| **Fear of COVID-19 (FCV-19)** | |
| α = 0.941; CR = 0.952; AVE = 0.740 | |
| FCV 1 | 0.886 |
| FCV 2 | 0.850 |
| FCV 3 | 0.831 |
| FCV 4 | 0.841 |
| FCV 5 | 0.867 |
| FCV 6 | 0.894 |
| FCV 7 | 0.851 |

α = Cronbach's Alpha Coefficient; CR = Composite Reliability; AVE = Average Variance Extracted; SFLs = Standardized Factor Loadings.

## 4. Results

Table 3 presented the results for discriminant validity with Fornel–Larker criterion, which stated that Average Variance Extracted (AVE) square root values should be more than the correlations among the variables. The bold diagonal values loaded for each variable

confirm that all the study variables discriminate from each other, and the AVE square roots are more than the standardized correlation. Discriminant validity is also confirmed by way of Henseler [59] through Heterotrait-Monotrait (HTMT) alues, and all the HTMT values are lower than one, which validates the discriminant validity. The mean values for mental health (4.563), perceived job insecurity (4.404), fear of COVID-19 (4.328), fear of economic crisis (4.156), and non-employability (4.154) are high. Whereas, all the standard deviation values are less than unity, which depicts that there is no issue of normality in the data for any of the study variables.

**Table 3.** Inter-construct correlation and discriminant validity.

| Constructs | Mean | SD | 1 | 2 | 3 | 4 | 5 |
|---|---|---|---|---|---|---|---|
| 1. MH | 4.563 | 0.544 | **0.857** | | | | |
| 2. FEC | 4.156 | 0.764 | 0.450 | **0.853** | | | |
| 3. NE | 4.154 | 0.601 | 0.373 | 0.494 | **0.878** | | |
| 4. PJI | 4.404 | 0.60 | 0.495 | 0.455 | 0.412 | **0.828** | |
| 5. FCV | 4.328 | 0.526 | 0.539 | 0.313 | 0.355 | 0.430 | **0.860** |

Note: MH = Mental health; FEC = Fear of economic crisis; NE = Non-employability; PJI = Perceived job insecurity; FCV = Fear of COVID-19.

We applied the structural equation model to test the hypothesized model with both moderating and mediating effect, as recommended by Hayes and Preacher [60]. This technique is suitable for the simultaneous analysis of all the study variables and followed a bootstrapping approach to evaluate the mediating impact of perceived job insecurity. The interaction terms (fear of economic crisis*fear of COVID-19) and (non-employability*fear of COVID-19) are calculated using the orthogonal interaction method to test the moderating effect of fear of COVID-19 on the link of fear of economic crisis–perceived job insecurity and non-employability–perceived job insecurity. The structural model is drawn on SmartPLS, and model fitness is also checked. Overall, the model is a good fit Standardized Root Mean Square Residual (SRMR) = 0.05; Chi-Square = 3339.71; Normed Fit Index (NFI) = 0.77). The R-square values for perceived job insecurity and mental health are 0.37 and 0.42, respectively, showing that the significant portion of the variance in these variables is explained by the independent variables.

Hypothesis1a posited that the fear of economic crisis would affect the employees' mental health through the mediation of perceived job insecurity. As stated in Table 4, fear of economic crisis was significantly and positively related with perceived job insecurity ($\beta = 0.247$; $p < 0.001$) and later, perceived job insecurity is positively related with the adverse mental health of the employees ($\beta = 0.225$; $p < 0.001$). Hypothesis1b stated that non-employability influenced the mental health of the employee through the mediation of perceived job insecurity, which is confirmed by Table 4, where non-employability is significantly and positively related with perceived job insecurity ($\beta = 0.189$; $p < 0.001$) and further, perceived job insecurity is positively associated with the negative consequences of the mental health ($\beta = 0.225$; $p < 0.001$). We also tested these mediation hypotheses directly through an indirect mediation model with bootstrap at a 95% confidence interval. The findings further validate the hypotheses H1a and H1b ($\beta = 0.055$; $p < 0.05$; $\beta = 0.042$; $p < 0.001$), respectively.

In Hypothesis 2a, we proposed that fear of COVID-19 moderates the indirect relationship of fear of economic crisis on mental health through perceived job insecurity. Table 4 confirmed that the interaction term (fear of economic crisis*fear of COVID-19) was significantly and positively associated with perceived job insecurity ($\beta = 0.178$; $p < 0.001$), which is further significantly affecting the mental health of the employees ($\beta = 0.225$; $p < 0.001$). Additionally, we directly tested the mediation model by linking the interaction term (fear of economic crisis*fear of COVID-19) with mental health through perceived job insecurity with bootstrap at a 95% confidence interval. The findings validate the results by indicating that the value of the indirect effect of mediation on the interaction term (fear of economic crisis*fear of COVID-

19) is significant on mental health (β = 0.267; *p* < 0.001) through perceived job insecurity, which provides support to Hypothesis 2a.

**Table 4.** Results for structural equation model.

| Hypothesized Path | Coefficients | *p* Values |
|---|---|---|
| **i. Direct Effect** | | |
| Fear of Economic Crisis → Perceived Job Insecurity | 0.247 | *** |
| Non-Employability → Perceived Job Insecurity | 0.189 | *** |
| Fear of Economic Crisis*Fear of COVID-19 → Perceived Job Insecurity | 0.178 | *** |
| Non-Employability*Fear of COVID-19 → Perceived Job Insecurity | −0.213 | ns |
| Perceived Job Insecurity → Mental Health | 0.225 | *** |
| **ii. Indirect Effect** | | |
| Fear of Economic Crisis → Perceived Job Insecurity → Mental Health | 0.055 | ** |
| Non-Employability → Perceived Job Insecurity → Mental Health | 0.042 | *** |
| **iii. Moderating Effect** | | |
| Fear of Economic Crisis*Fear of COVID-19 → Perceived Job Insecurity → Mental Health | 0.040 | *** |
| Non-Employability*Fear of COVID-19 → Perceived Job Insecurity → Mental Health | −0.048 | ns |

Note: *** < 0.001; ** < 0.01; * < 0.05.

We plotted two different interaction graphs to explain the moderation results. Figure 2 explained the moderating role of fear of COVID-19 between fear of economic crisis and perceived job insecurity. The graph shows that the relationship between fear of economic crisis and perceived job insecurity is stronger under a high fear of COVID-19. The notable point on the moderation graph is perceived job insecurity at a high and low level of fear of COVID-19. The relationship between fear of economic crisis and perceived job insecurity is stronger when COVID-19 fear is high (see Figure 2), meaning when fear of economic crisis and fear of COVID-19 are consistent. This eventually supports Hypothesis 2a.

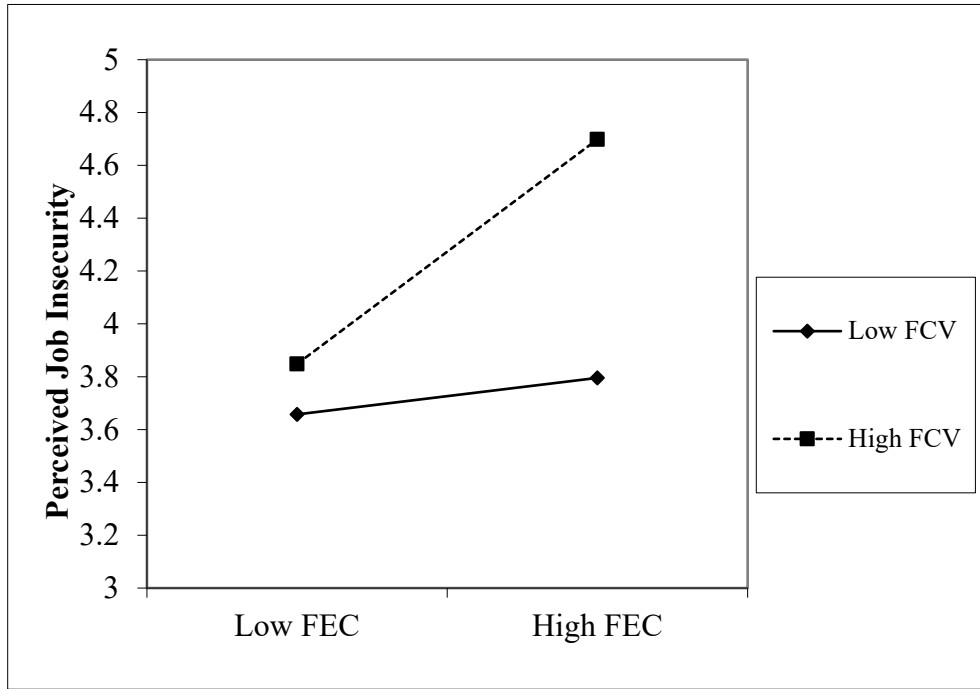

**Figure 2.** Moderation of FCV-19 between fear of economic crisis and perceived job insecurity.

Furthermore, we plotted the conditional indirect effects for mental health on three different moderator levels with 95% upper and lower bootstrap confidence intervals by following the recommendations of Hayes [61], as presented in Figure 3. Figure 3 shows the indirect effect of the interaction term (fear of economic crisis*fear of COVID-19) on mental health through perceived job insecurity at higher (+1SD) and lower (-1 SD) levels from the mean values of fear of COVID-19. The value of the indirect effect is significant if zero is excluded from the upper and lower confidence interval limit. The total indirect effect is conditional on the moderating variable (fear of COVID-19) at the mean (4.33), at 1 SD above the mean (4.78), and 1 SD below the mean (3.88). These findings are also consistent with the proposed Hypothesis 2a.

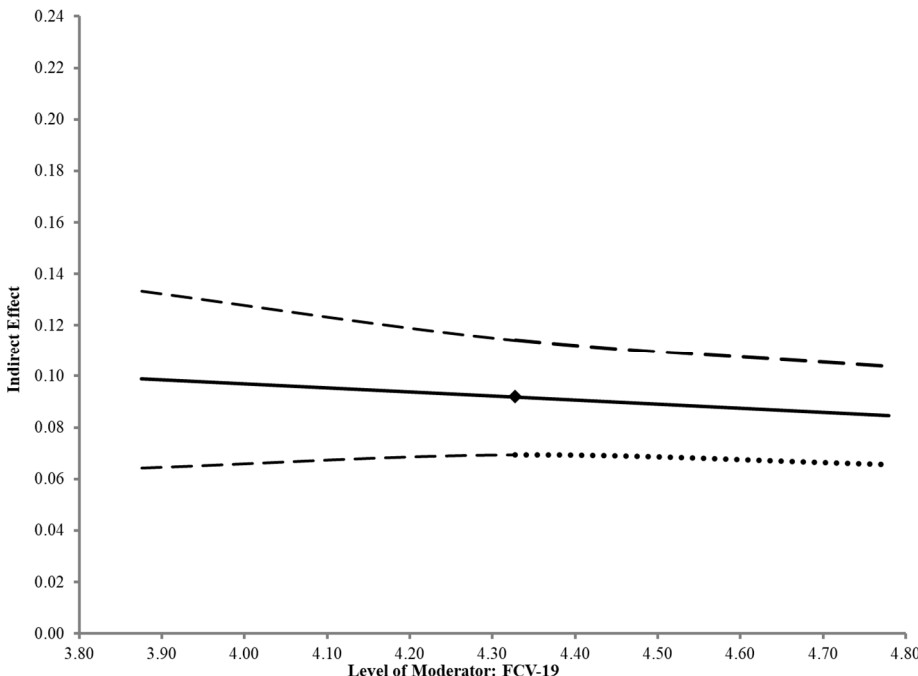

**Figure 3.** Effect of fear of economic crisis on perceived job insecurity through mental health at levels of FCV-19.

## 5. Discussion

The present study investigated the effect of fear of economic crisis and non-employability on employees' mental health through perceived job insecurity. It also analyzes the moderating role of fear of COVID-19 between direct and indirect relationships. This study empirically examines the underlying framework by conducting an online survey from 372 employees of the hospitality industry. Data collection were initiated during the period of the COVID-19 pandemic (May to July 2020). Questionnaires were distributed to the targeted sample through online platforms. The hypothesized model was tested through a structural equation model. This study has three significant findings. First, it confirms the direct relationships between fear of economic crisis, non-employability, fear of COVID-19, perceived job insecurity, and employee mental health. The prior literature supports all these findings.

Previous studies reported that economic instability and financial crisis are the leading cause of depression, stress, and anxiety among the employees [10]. Researchers consider economic and political unrest as a major cause of psychological diseases that adversely affect the health of the person. Non-employability threats also lead toward adverse health consequences [32]. Employees' perception regarding their working competencies that do not allow them to acquire another job created a constant stress, which negatively influences their mental health [23]. Bao [48] claimed that high fear of COVID-19 is now becoming the root cause of mental stress among the employees. The present research also supports the

idea presented by De Witte [12] and Caroli and Godard [62] claiming that the perceived job insecurity has a positive impact on the mental issues of the employees. An employee cannot function properly if they are psychologically stressed and feel that their job is not secure [9].

Second, economic stability is an essential factor in the growth of any sector, and the economic crisis is a state when employees are adversely affected and creates a perception that their job will not be secured. In line with this notion, the present study has also observed a positive relationship between fear of economic crisis on perceived job insecurity, which confirms the findings of Kalleberg [37]. Previous studies consider employability a significant predictor of job insecurity, and low employable employees are more likely to perceive job insecurity [31]. Our findings are also in line with the previous studies. These results validate the mediating role of perceived job insecurity between fear of economic crisis and employee mental health. Similarly, employability helps the employees feel more secure about their jobs even during the crisis period [54]; however, non-employability increases the perceived job insecurity that adversely affects the employee mental health.

Thirdly, the idea that makes this research unique is to test the moderating role of fear of COVID-19 between direct and indirect relationships. The present study also concludes that the fear of COVID-19 moderates the direct connection of fear of economic crisis on mental health, supporting the results of French [47]. The effect of fear of economic crisis on mental health through perceived job insecurity is also stronger when employees have a higher fear of COVID-19. Due to the pandemic, the fear of perceived job insecurity has increased [42], thus adversely affecting the employees' mental health [43]. This situation motivates the scholars, policy-makers, and social scientists to explore the in-depth relationship of fear of economic crisis, perceived job insecurity and employees mental health. In line with explaining the moderating role of fear of COVID-19, we have plotted the two different interaction graphs that comprehend the effect of moderating variable in a better way. Figure 2 explained that the relationship of fear of economic crisis and perceived job insecurity strengthened when fear of COVID-19 is high among the employees. Figure 3 also validates the effect of the interaction term (fear of economic crisis*fear of COVID-19) on mental health through perceived job insecurity.

In the end, we conclude that the fear of economic crisis and non-employability enhances the perceived job insecurity among the employees and is becoming the root cause for the various psychological problems such as stress, depression, anxiety, and uncertainty during COVID-19 outbreak. It also increases the attention of the management toward the adoption of technological skills and innovative ideas that lead toward a new era of digital development [63]. This new advancement suggests the utilization of robotic technology and suggests the application of artificial intelligence that transforms the existing structure of the hospitality industry, which is in fact the need of the current situation of COVID-19. Finally, this study discusses the implications for the scholars, management, and employees, addresses the limitations, and provides future directions to the researchers in the next section.

*Open Innovation in Hospitality Industry*

Innovation is considered as a source for gaining the competitive advantage for the organizations [64]. It is also a necessity to improve the performance of the companies, especially for the service sector. The present research highlighted the importance of innovation for the survival of hospitality industry in the current age. It is believed that the hospitality industry may have to switch toward innovative business strategies rather than a traditional business model to gain the trust of its customers and provide them with a safe experience. Although the world was already shifting toward e-business and e-commerce, this pandemic has further forced the development of open innovation in the hospitality industry. Open innovation in the hospitality industry has been an imminent field of advancement [65]. Research explicitly states the importance of open innovation in

the manufacturing and services sector, but there is clear evidence of a lack of research in the hospitality sector [66].

The current study focused on the impact of fear of economic crisis and non-employability on the perceived job insecurity and eventually on the mental health of the employees during COVID-19 in the hospitality sector of Pakistan. One of the major implications drawn from the results was that this pandemic has done much harm to this industry. Customers will take time to gradually build up trust and start their routine with outdoor activities. Hence, the hospitality sector should be ready, and if this sector wants to survive and again gain a competitive edge, then it needs to adapt to the open innovation and provide customers with an environment that is safe and has less human interaction.

## 6. Conclusions

The current study sheds light on how the fear of COVID-19 can impact the mental health of the employees of the hospitality industry by collecting data from 372 employees. It further empirically investigated the effect of fear of economic crisis and non-employability on mental health through perceived job insecurity. Results show that perceived job insecurity mediates the relationship of fear of economic crisis, non-employability, and mental health. However, the moderating role of fear of COVID-19 is confirmed only in the direct relationship of fear of economic crisis and mental health and also on its indirect path through perceived job insecurity. Our findings provide a guideline to the management of hospitality industry on how to deal with the employee mental health issues by coping with the fear of crisis that is important for designing the strategies of effective employee management. It also encourages the industry to adopt technological infrastructure for maximizing their profitability in the present era of advancement.

### 6.1. Implications

The present study enriches the existing knowledge in the field of behavioral studies. It also suggests many practical implications for the management of the hospitality industry to cope with the employee mental health issues and emphasizes the adoption of technological infrastructure to enhance their revenues. The significant implications of the study are as follows. Firstly, it contributes to the existing literature of employee's well-being by examining the influence of macro-level factors such as fear of economic crisis and non-employability on the micro-level psychological factors such as perceived job insecurity and mental health of the employees in the hospitality industry of Pakistan. Secondly, it will guide the organizations to manage their employees during the crisis period. It highlighted the need to understand the psychological factors and emphasized that if management wanted to increase the employee's performance, they must give importance to the psychological aspects, stimulate optimism, and create a positive atmosphere to promote employee well-being.

Thirdly, from a practical perspective, the focus of the employers is usually on profit generation; however, specifically in the hospitality industry, where the employees have to interact with customers and provide them with services, then the mental health of employees should be given priority. Fourthly, this research might help the organizations deal with the employees' non-employability threat by providing them with training and keeping them up to date with advanced knowledge practices. Finally, this study highlighted the need for digital infrastructure investment in the hospitality industry. Consumer behavior will be expected to change after the COVID period. Consumers will prefer less in-person interaction in hotels with greater hygiene standards. These customers' expectations will eventually lead toward a structural shift, where investment in technology will become a necessity not for engaging the customers but also for the well-being of the employees.

### 6.2. Limitations and Future Recommendations

This study has faced certain limitations that are important to incorporate to enhance the research scope. First, we have collected the data from the employees of the hospitality

industry only, making it difficult to generalize the findings on other industries. Second, a limitation of the study is that it takes into account only cross-sectional design, because the study explores relationships around which scholars are still discussing to establish their direction. For instance, some studies confirmed that employability predicts job insecurity, but some others found job insecurity to predict employability (see: Cuyper, Broeck [67]). Similarly, the initial level of mental health could impact the fears of economic crisis, and COVID-19 is still debatable. These guidelines may open new horizons for the future researcher so that they might be considered in the future.

Third, in this research, perceived job insecurity is considered as a mediator; however, it is suggested to the future researchers to use a different mediator such as scarce social support, job stress, and future career anxiety in the same model to get a better understanding of the relationship between fear of economic crisis, non-employability, and mental health. The fourth limitation can be the transition in the conditions, at the time of preparing this article. Since data were collected when fear of COVID-19 was at a peak, other related conditions such as lockdown and quarantine prevailed at many places. Thus, prior literature shows that a high fear level may lead to a different outcome [68]. Previous studies also highlight that such fear increases the social and professional trauma, leaving employees with no other choice than to face financial and mental stress [69].

Fifth, fear of COVID-19 is a new construct in the literature. Therefore, it is suggested that future researchers should test our model in the context of other countries to validate or contradict the current results. Furthermore, it is proposed to conduct a comparative study and investigate the effect of fear of COVID-19 across diversified age groups, gender, and employment status. Sixth, this study only discusses the economic and psychological factors associated with employees' mental health. It does not cover the organizational perspective during the crisis period, which strategies organizations are adopting to cope with the situation by satisfying employee needs and enhancing profitability; that area is open for the future researchers for further investigation. Finally, this research is based on the self-reported questionnaire collected at one point in time; therefore, the generalizability of the findings might be questionable. That is why it is recommended that the future scholar use a longitudinal research design to confirm the current results' consistency.

**Author Contributions:** Conceptualization, K.I.K. and A.N. (Adeel Nasir); methodology, M.I.K. and A.N. (Amna Niazi); software, K.I.K. and M.I.K.; validation, M.H., A.N. (Amna Niazi), and A.N. (Adeel Nasir); formal analysis, K.I.K.; investigation, M.I.K.; resources, M.H. and A.N. (Adeel Nasir); data curation, M.I.K. and A.N. (Amna Niazi); writing—original draft preparation, K.I.K., A.N. (Amna Niazi), and A.N. (Adeel Nasir); writing—review and ed-iting, M.H., and M.I.K.; visualization, A.N. (Adeel Nasir); project administration and supervi-sion, K.I.K. All authors have read and agreed to the published version of the manuscript.

**Funding:** This research received no external funding.

**Informed Consent Statement:** This study was carried out through the approval of the Departmental Committee of Professional Ethics FAST School of Management, National University Lahore Campus, Pakistan, in accordance with the Declaration of Helsinki. Informed consent was obtained from all individual participants included in this study.

**Data Availability Statement:** Data is available, and can be provided on request.

**Conflicts of Interest:** The authors declare no conflict of interest.

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
