# Peer review of "The Effect of COVID-19 on the Hospitality Industry: The Implication for Open Innovation"

_2199-8531, doi:10.3390/joitmc7010030_

Round 1
Reviewer 1 Report
I was very curious to read this paper because I think it focuses on the issue that is drammatically relevant. However, I think that the quality of paper needs to be improved significantly. I found the literature review quite poor and it is not enough to support the formulation of the hypotheses.
In addition, I think that the study presents some relevant limitations that need to be underline at least.
Finally, I think that the authors should put more effort to explain the theoretical contribution of their work.
Below I list my suggestions, hoping they may be useful to improve the paper.
Introduction
-Lines 100-124: A clear definition of “fear of economic crisis” is missing.
Between lines 100 and 108 I found some not really focused affirmitions about fear, that are in some cases questionable too (for instance, I read with a little of disappointment “The factor fear has not only been considered as a strategy to motivate employees in the corporate settings”).
In addition, the authors seem to use interchangeably the terms “fear of economic crisis” and “economic stress”, but they are not the same construct. Given that the authors used the scale of Giorgi et al. (2013) to measure the fear of economic crisis, they could use the same definiton of fear of economic crisis these authors purpose (see for instance, Giorgi et al, 2020).
- Lines 125-140: A clear definition of “non-employability” is missing.
According to Giorgi et al (2013, 2020), non-employability could be introduced as a dimension of econimic stress.
In any case, it is necessary to define what it means in terms of self-perception of personal skills and ability to gain and mantain a job.
In addition, it is necessary to cite more than a very old reference (Kessler, House, and Turner, 1987) to assume a relationship between Non-employability and Mental Health
- Lines 140-167: The authors ignored the amount of literature that explored the relationship between perceived employability, job insecurity and wellbeing (i.e. Silla et al 2009, Otterbach and Sousa-Poza, 2016). It would be very important to strenght the justification of H1
- Lines 168-230: A clear definition of “fear of COVID 19” is missing.
Reading this paragraph it seems that the fear of COVID 19 refers only to the fear of negative economic impact, but it is incorrect. Describing the measures they used, the authors cite an item “I am afraid of losing my life because of coronavirus-19”, and it confirms that the fear of COVID 19 does not refer only to negative effect on econimic situation, but also on health.
Method:
Some additional information and clarifications are necessary:
- Participants: how many hotels were contacted?
- Measures: it is necessary to clarify if and how the authors translate some scales. In addition, some describtions of the scale used are not clear: the auhtors used an unidimensional measure of perceived job insecurity so it is not clear why they decided to add three items “from Ashford, Lee [49] and Greenhalgh and Rosenblatt [31] that explained the multidimensional approach” (lines 256-257); General Health Questionnaire is an unidimensional scale so I did not understand why authors wrote “Mental health is a three-dimensional scale that includes anxiety & depression, social dysfunction and loss of confidence” (lines 266-267)
Results:
-the labels “fear of economic crisis*fear of covid 19” and “no-employability*fear of covid 19” for the interaction terms are preferable.
-the describtion of figure 3 (lines 338-339) is incorrect.
Discussion:
-the theoretical implications should be discussed in the light of previous studies. What is confirmed? What is new? The authors should comment the results step by step. For instance, how can the authors explain the fact that fear of covid interact with fear of economic crisis but not with non-employability in determing perceived job insecurity.
Limitation
The cross-sectional design is a strong limitation especially because the study explores relationships around which scholars are still discussing to establish their direction. For instance, some studies confirmed that employability predicts job insecurity but some others found that it is the job insecurity to predict employability (i.e. Nele De Cuyper & Anja Van den Broeck & Hans De Witte, 2015). Similarly, the initial level of mental health could impact on the fears (of economic crisis and covid 19). Consequently, I suggest to the authors to underline this limitation in the conclusion paragraph (currently, it is just mentioned)
Author Response
"Please see the attachment."

Reviewer 2 Report
The present manuscript aimed to investigate the effect of the economic crisis and non-employability on the mental health of employees.
First of all, I would like to suggest to the authors to improve the abstract of the manuscript. I mean that I would like to see a balance between the introduction/methods and results/conclusion. More specifically, I believe that you should provide more details in terms of results. In fact, the abstract should attract potential readers and it needs more information.
Secondly, I ask you to provide more details of chapter 2 "Literature review and Hypothesis Development". I would like to know how you carried out the search and what archives you analyzes.
Furthermore, pay attention to figure 1. I noticed that it is not mentioned in the main text. As you should know, before adding a figure, you have to cite it in the text.
Regarding table 1, please add a column with the % value.
Finally, I suggest sending the article to a proofreading service to improve the English.
Author Response
"Please see the attachment."

Reviewer 3 Report
This is a suvey studying job insecurity, fear of non-employability and of economic crisis via SEM in 372 hotel workers.
It is a potentially interesting study but there are a few substance and formal issues
1) the introduction and discussion section are full of truisms, common sense statements and discursive language not suitable to a scientific publication. English language also needs professional editing
Examples:
COVID-19 has made the world economy paranoid that leads towards severe 6 socio-economic crisis and sustainable psychological distress
pandemic COVID-19 has emerged as significant health and 28 psychological resilience worldwide
In the private sector, the fear of job insecurity is present from the start, but during the recent 42 period of COVID- 19, the fear has reached a higher level
The fear of 101 the unknown is a driving force to take advantage of the insecurities of the other party.
2)on the other hand there is very little information of the hospitality industry and how much has been affected by the COVID pandemic. We are told that people were dismissed but authors should be far more specific
3) Title is not informative enough: hospitality industry is not mentioned nor place
4) no ethical committee approval mentioned
5) most references are not complete
6) see for background: Int. J. Environ. Res. Public Health 2020, 17, 7366; doi:10.3390/ijerph17207366
Author Response
"Please see the attachment."

Round 2
Reviewer 1 Report
I would like to thank the authors to have taken into account my suggestions. I found that the paper has been really improved.
I have only two additional request of minor revisions that are the following:
- Lines 168-169: “Literature is filled with the evidence that proves a negative relationship between non-employability and perceived job insecurity [41]”.
Are you sure that the relationship between non-employability and perceived job insecurity is negative? Maybe, you would have written “the relationship between employability and perceived job insecurity is negative”. In addition, I am not sure that the citation 41 deals with it (Astell- Burt and Feng, 2013)
- The authors wrote “The labels "fear of economic crisis*fear of covid 19" and "non-employability*fear of covid 19" for the interaction terms are used as per the suggestion of reviewer” but I did not find this modification in the Result paragraph (see lines 307, 329, 332, 334 and Table 4)
Author Response
Point 1: Lines 168-169: “Literature is filled with the evidence that proves a negative relationship between non-employability and perceived job insecurity [41]”. Are you sure that the relationship between non-employability and perceived job insecurity is negative? Maybe, you would have written “the relationship between employability and perceived job insecurity is negative”. In addition, I am not sure that the citation 41 deals with it (Astell- Burt and Feng, 2013)
Response 1: As suggested by the reviewer the citation has been changed to more relevant one (mentioned below) and the mistake of writing non- employability with employability has been rectified.
Cuyper, N. D., Bernhard‐Oettel, C., Berntson, E., Witte, H. D., & Alarco, B. (2008). Employability and employees’ well‐being: Mediation by job insecurity 1. Applied Psychology, 57(3), 488-509.
Point 2: The authors wrote “The labels "fear of economic crisis*fear of covid 19" and "non-employability*fear of covid 19" for the interaction terms are used as per the suggestion of reviewer” but I did not find this modification in the Result paragraph (see lines 307, 329, 332, 334 and Table 4)
Response 2: Thank you reviewer 1 for highlighting the mistake, now we have rectified it throughout the document.
Reviewer 3 Report
Manuscript is clearly improved in the revised version and Authors have addressed all issues raised.
The lack of an Ethical Committee or Institutional Review Board of the research is in my opinion a major problem, considering also that the research involved the use of rating sclales used to screen for mental disorders as the Author acknowledge. To my knowledge in most Countries this would call for at least Institutional Review Board approval.
Author Response
Point 1: The lack of an Ethical Committee or Institutional Review Board of the research is in my opinion a major problem, considering also that the research involved the use of rating sclales used to screen for mental disorders as the Author acknowledge. To my knowledge in most Countries this would call for at least Institutional Review Board approval.
Response 1: We have taken the prior approval from the institutional ethical committee before initiating the research (See attached letter). The management and employees were also informed about it. The management and employees who participated in this study were also informed about the aim and scope of the study. They were ensured that their responses were only used for research purpose, and the results will be shared with them for the purpose of recommending them with policies and practices that may help them in their professional career.

Round 3
Reviewer 3 Report
It is unclear why Intitutional Review Board approval was not mentioned before.
I am happy that this has been obtained, it should be mentioned also in the Methods Section of the paper
Author Response

(The authors gave the same response as above.)
